# Thyroid Cancer Risk Factors in Children with Thyroid Nodules: A One-Center Study

**DOI:** 10.3390/jcm10194455

**Published:** 2021-09-28

**Authors:** Iwona Ben-Skowronek, Joanna Sieniawska, Emilia Pach, Wiktoria Wrobel, Anna Skowronek, Zaklina Tomczyk, Anna Mlodawska, Magdalena Makuch, Magdalena Malka, Czeslaw Cielecki, Pawel Nachulewicz

**Affiliations:** 1Department of Pediatric Endocrinology and Diabetology with the Endocrinology and Metabolic Laboratory, Medical University of Lublin, 20-093 Lublin, Poland; joanna.sieniawska@gmail.com (J.S.); emilia.k.pach@gmail.com (E.P.); WiktoriaKW_97@interia.pl (W.W.); ania.skowron97@gmail.com (A.S.); zaklina.tomczyk96@gmail.com (Z.T.); assadana6@gmail.com (A.M.); m.makuch8@onet.pl (M.M.); magdalena.malka@wp.pl (M.M.); 2Department of Paediatric Surgery and Traumatology, Medical University of Lublin, 20-093 Lublin, Poland; czeslawcielecki@gmail.com (C.C.); nachulewicz@msn.com (P.N.)

**Keywords:** thyroid nodules, thyroid cancer, risk factors, ultrasound, children

## Abstract

Thyroid nodules are common in the adult population (13%), but in childhood, they are relatively rarely diagnosed (0.2–5%). The risk factors and diagnostic and therapeutic algorithms are well-known and effectively used in adults, but no clear procedures supported by scientific research are available in the pediatric population. Our aim in this study was to identify predictive factors for thyroid cancer in a pediatric population. We retrospectively analyzed 112 children (80 girls and 32 boys, aged 0.6–18 years, with an average group age of 13.4 ± 4.5 years) with thyroid nodules who presented or were referred between 2010 and 2021. A total of 37 children qualified for partial or total thyroidectomy. After histopathological nodule examination, the most common cases were benign lesions in 23 patients (57.5%) and malignant lesions in 14 children (32.5%). Solitary benign thyroid nodules were found in 16 children (40%). Malignancy risk was higher in children with increased nodule diameter (greater than 7 mm; *p* = 0.018) or hypoechogenic lesions in ultrasound (*p* = 0.010), with no correlation between increased blood flow in the vessels and tumor diagnosis. The relative risk of developing thyroid cancer for class III was found to be higher in comparison to adults and 11.1 times higher than for classes I and II combined.

## 1. Introduction

### 1.1. Thyroid Cancer Risk Factors

Nodular goiter is a rare condition in children and adolescents, with a prevalence of 0.2–5% [1]. However, nodules have a higher risk of malignancy (25%) in these age groups than in adults [2]. The risk factors for thyroid nodules are female sex, a positive family history of thyroid disease (e.g., familiar multinodular goiter or familiar nonmedullary thyroid cancer), previous thyroid disease, chronic thyroiditis, autoimmune disease, and taking medication with hormones and steroids [2,3,4,5,6]. Moreover, an elevated thyroid-stimulating hormone (TSH) level and dysfunction of the TSH receptor, sometimes connected to iodine deficiency, can cause thyroid nodules [2,4]. The most important risk factor is exposure to radiation, especially the head and neck, including radiotherapy (e.g., Hodgkin’s lymphoma treatment and hematopoietic stem cell transplantation) [2].

A significant increase in the incidence of papillary thyroid cancer was noticed in the areas covered by radiation after the explosion of the Chernobyl Nuclear Power Plant in 1986 and Fukushima in 2011 [7]. Some sources report that nodules with microcalcifications, hypoechogenic nodules, intranodular vascularization, and elevated TSH levels increase the risk of thyroid cancer in children [8]. Additionally, thyroid nodules are more often observed in several genetic syndromes such as Cowden syndrome, phosphatase and tensin homolog (PTEN) mutation, McCune–Albright syndrome, Peutz–Jeghers syndrome, Li–Fraumeni syndrome, or Beckwith–Wiedemann syndrome [2,4,5,9,10].

### 1.2. Molecular Aspects of Thyroid Cancer

Thyroid cancer in adults is more locally invasive than in children. Various gene mutations can be found in papillary cancer. For adults, the most common genetic changes are B-Raf proto-oncogene (BRAF V600E) and RAS (a family of small membrane GTPases) point mutations and rearranged in transformation/papillary thyroid carcinomas (RET/PTC) fusions. Human telomerase reverse transcriptase (HTERT) promoter mutations have been observed in adult papillary thyroid cancer (PTCs) and are associated with a more aggressive phenotype [11,12,13,14,15,16]. Gene mutations and RET/PTC, ETV6-NTRK3, and BRAF fusions (AGK-BRAF and AKAP9-BRAF) are more common in the pediatric population, especially up to the age of 10 years. Research has shown that 50% of cancers have some kind of gene rearrangement, regardless of radiation exposure [12,13]. The younger the age, the more important fusion oncogenes are in the development of PTC [17]. Medullary thyroid carcinoma occurs in 25% of the dominant component of hereditary multiple endocrine neoplasia type 2 (MEN2); mutations of the proto-oncogene RET are the most frequently involved in cancer pathogenesis, but there are many other mutation patterns [13,18].

### 1.3. Physical Examination

In most pediatric patients, the disease is asymptomatic when thyroid nodules are detected. Signs of compression such as dyspnea, dysphagia, dysphonia, hoarseness, and pressure or pain in the neck area are extremely rarely observed; if they occur, they indicate local advancement [19].

Attention should also be paid to the symptoms of hyperthyroidism in the patient. The nodules are most often found accidentally by the patients themselves or during a standard physical examination by a doctor [20]. With both viewing and palpation, the first required action is to assess the size and symmetry of the thyroid gland. Physical examination should also focus on the cervical lymph nodes, especially in regions II, III, IV, V, and VI. If nodules are found, it is important to determine their number, approximate size, texture, soreness, and displacement [4].

Oncological anxiety can be caused, especially by hard and non-movable lesions; however, even the smallest palpable nodule should be subjected to further diagnostics [19]. Most lesions smaller than 1 cm are imperceptible unless they are superficial. Nodules on the posterior wall of the thyroid gland are particularly difficult to access for examination, and even those of considerable size can be easily missed.

### 1.4. Diagnostics

#### 1.4.1. The Thyroid Gland in Ultrasound Images

Due to the intensive technical development of ultrasound devices, we are able to detect even small changes (1–2 mm in diameter) and obtain information about the exact location, echogenicity of nodules, the condition of the surrounding lymph nodes, and the presence of the ultrasonographic features of increased risk of malignancy (Table 1) [21,22,23,24,25,26].

The patients can qualify for FNAB according to EU-TIRADS classification; however, EU-TIRADS is performed for adult patients but is disputable in children.

Another difference in the pediatric population is the recommendation that all FNABs (fine needle aspiration biopsies) should be performed under ultrasonography guidance. This option is preferred because of the more malignant changes in this age group and because it allows avoiding the difficulty in collaboration with a young patient, with the possible result of nondiagnostic testing and the need for repeated examination [27]. Additionally, the performed FNAB changes the ultrasound image of the lesion, which could adversely affect further imaging observation [5].

Other imaging tests, such as radiography, computed tomography (CT), magnetic resonance imaging (MRI), or positron emission tomography (PET), do not provide any advantage over ultrasound examination, but their use in looking for metastatic foci may be considered [28,29].

#### 1.4.2. Hormonal and Immunological Diagnostics

An indispensable element of diagnosing focal lesions in the thyroid gland is laboratory diagnostics. In the case of nodular goiter, the level of TSH should always be determined, bearing in mind that TSH > 2.5 uIU/mL is an independent predictor of pediatric thyroid cancer [30].

In the case of lowered TSH levels (or near the lower limit of normal), 99mTc thyroid scintigraphy should also be performed. However, over the past few years, the significance of this test has decreased significantly [20]. Additionally, anti-thyroid peroxidase (TPO) or other antibodies could be marked, calcitonin should be determined in case of suspected medullary carcinoma, RET (gene of tyrosine kinase receptor) mutation could be confirmed to exclude medullary carcinoma before planned surgery, or elastography could be performed before the planned FNAB to select the correct puncture site [29].

### 1.5. FNAB Results and the Choice of Further Treatment

According to the 2015 recommendations of the American Thyroid Association (ATA) regarding the management of children with thyroid nodules and differentiated thyroid cancer, the ultrasound-guided fine needle aspiration biopsy (US-FNAB) result is a decisive factor for further treatment.

#### 1.5.1. Histopathological Diagnosis and Further Management

The treatment of thyroid tumors depends on the histological type. Among the pediatric population, PTC is the most common, followed by follicular thyroid cancer (FTC), medullary thyroid cancer (MTC), and anaplastic cancer, among others [6].

##### Benign Thyroid Nodule

In the case of benign thyroid nodules, the 2015 ATA guidelines recommend repeating the ultrasound in 6–12 months and reassessment. Surgical procedures, e.g., lobectomy, should be included if, due to the size of the nodule, compressive symptoms are present or because of cosmetic reasons or the patient’s/parents’ desire. If the nodule is stable in the reassessment, the next US should be repeated every 1–2 years, but if the nodule is growing or other suspicious symptoms are present in the US, FNAB or surgery should be performed. Levothyroxine suppressive therapy reduces the nodule size and decreases the risk of new lesions, but data about the long-term safety profile of this treatment method are ambiguous. If a pediatric patient has an autonomous thyroid nodule, considering the risk of DTC, which is incidentally found in up to one-third of patients with autonomous thyroid nodules, surgical resection is the procedure of choice. There is no information available on the safety and efficacy of other treatments in the pediatric population. In the absence of clinical symptoms, surgery can be postponed by performing FNAB if the nodule has PTC features [5].

##### Malignant or Suspicious Lesions

An FNAB result indicating a malignant or suspicious lesion is an indication for a surgical procedure, most often total thyroidectomy (TT), which is excision of the right and left lobe, pyramidal lobe (if present), and isthmus. If the nodule is benign after histopathological examination, the level of thyroid hormones should be monitored, and a clinical evaluation of the patient should be performed; if a malignant change is demonstrated, the procedure depends on the type of neoplasm [5].

When PTC is detected in a patient, TT is usually recommended. For a small, unilateral tumor, near-TT may be considered, leaving < 1–2% of the thyroid tissue around the recurrent laryngeal nerve and/or upper parathyroid glands to avoid damage to these structures [5]. The procedure after FTC detection does not differ from PTC treatment: the recommended treatment is lobectomy with isthmusectomy. In the case of a tumor larger than 4 cm infiltrating blood vessels, TT is recommended due to the higher risk of metastasis [5].

According to the 2015 ATA guidelines, the FNAB result, meaning MTC with a calcitonin level below 500 pg/mL, is an indication for TT, in some cases in combination with external beam radiotherapy. In the case of calcitonin levels above 500 pg/mL, the presence of distant metastases should be assessed; if they are absent, proceed as in the case of calcitonin levels below 500 pg/mL, but if they are present, TT should be performed, regional disease treated, and systemic therapy started with drugs with thyroid kinase inhibitor (TKI) groups [31].

Prophylactic TT is recommended in patients with MTC who have an RET germline mutation and the resulting multiple endocrine neoplasia type 2A (MEN2A) or multiple endocrine neoplasia type 2A (MEN2B) syndrome [31].

#### 1.5.2. Assessment with TBSRTC and Further Treatment

FNAB results are reported according to the Bethesda classification, and the ATA guidelines recommend the use of the same system for describing pediatric thyroid cytology as that used for adults [32].

The risk of thyroid nodule malignancy estimated by the FNAB score and the Bethesda classification plays a key role in selecting further treatment. In the adult population, including NIFTP for malignant tumors, class I (unsatisfactory or non-diagnostic biopsy) is associated with a malignancy risk of 5–10%, class II (benign lesion) with a risk of 0–3%, class III (AUS/FLUS) with a risk of 6–30%, class IV (FN or SFN) with a risk of 10–40%, class V (suspicious for malignancy) with a risk of 45–75%, and class VI (malignant) with a risk of 94–99% [33].

If the biopsy result is unsatisfactory or non-diagnostic, which means class I Bethesda, US and FNAB should be repeated within 3–6 months. When the nodule is stable in the examinations and the FNAB result shows a benign lesion, ultrasound follow-up is recommended for 6–12 months, whereas evidence of abnormalities in these tests is an indication for surgery [5].

In the case of Bethesda class II (benign lesion), clinical and ultrasound follow-up is recommended [33].

The management of the TBSRTC III category, AUS/FLUS, is not as obvious as for the borderline classes of this system due to the intermediate risk of malignancy and the associated diagnostic uncertainty, along with difficulties in predicting the patient’s prognosis. Management suggestions include replication of FNAB, molecular testing, or lobectomy, depending on the clinical condition [33].

A similar problem also occurs with class IV, which describes a follicular neoplasm or suspicious follicular neoplasm. The risk of malignancy of these lesions is so high that choosing the right treatment is a priority. Surgical procedures (hemithyroidectomy or lobectomy) supplemented with molecular testing are usually recommended [33].

In class V, in the case of lesions suspected of being malignant, and in class VI (malignant nodules), the procedure is usually an operation: lobectomy or near-TT [33].

The above practices for each class of TBSRTC apply to adults but should be adapted for the assessment of pediatric patients.

## 2. Patients, Materials and Methods

A retrospective analysis of 112 patients with a thyroid nodule who presented or were referred to our center between 2010 and 2021 was performed. The medical records of patients including preoperative, intraoperative, and postoperative data were reviewed. We analyzed the following information: age, sex, attending illness, clinical symptoms, physical examination, family history, thyroid ultrasonography, FNAB result, surgical procedures, histopathological findings, complications, and follow-up findings. The diagnostic pathway for patients with thyroid nodules is shown in Figure 1.

In all patients, TSH, free thyroxin (Elisa ABBOTT), thyroid peroxidase autoantibodies (TPO Ab), and thyroglobulin autoantibodies (Tg Ab; Elisa DAKO, Copenhagen, Denmark) were determined annually. Hashimoto’s thyroiditis was diagnosed in children with positive tests for TPO Ab and Tg Ab at normal or decreased free thyroxin levels.

Autoantibodies were analyzed in children with decreased TSH and increased free thyroxin; elevated TPO Ab and Tg Ab indicated Hashitoxicosis, a form of Hashimoto’s thyroiditis. Elevated thyroid stimulating immunoglobulin (TSI) antibodies (RIA BRAHMS, Berli, Germany) were used to diagnose Graves’ disease. In all patients, the calcitonin level in serum was determined.

Thyroid ultrasonography was performed using a Siemens Sonoline Elegra. High-resolution linear transducers (7.5 MHz) were used. The protocol includes two-dimensional (2D) gray-scale imaging of the right and left lobes, isthmus, and, if present, the pyramidal lobe, with sagittal and transverse views. The thyroid sonographic reports included the thyroid’s position, shape, size, content, echogenicity, and vascular pattern. Three linear dimensions were measured for each lobe: the length and anterior–posterior diameters of each lobe in the sagittal view, and the width in the transverse view. The volume of each lobe can be automatically calculated after recording the abovementioned three linear dimensions using the ellipsoid equation with a correction factor: V (volume in mL) = L (length in cm) × A-P (anteroposterior diameter in cm) × W (width in cm) × 0.523. The nodule dimensions were measured: length, height, and width. Every thyroid nodule was measured and assessed individually, including its location, size, internal structure, and vascularity by color Doppler.

### Statistical Methods

To verify the hypotheses for qualitative variables, a chi-squared test, a chi-squared test with Yates’s correction, or Fisher’s exact test was performed, as appropriate. In the case of quantitative variables, due to the patient groups (group with a histopathological result of 0 and a histopathological result of 1) being asymmetrical, the non-parametric Mann–Whitney U-test was used to check for significant statistical differences between the groups. The level of statistical significance was set to *p* = 0.05. For variables such as length, height, width, and volume of tumor, ROC curves were used to assess the quality of the quantifiers. The value of the AUC and the optimal cut-off point were determined in each case.

Statistical analyses were conducted using the Statistica 13.3 program.

Before commencement of the investigations, all parents signed an informed consent form. The investigation was accepted by the local Ethics Committee at the Medical University in Lublin.

## 3. Results

Retrospectively, we investigated 112 children (80 girls and 32 boys) diagnosed with thyroid nodule in our center between 2010 and 2021. Visible or palpable swelling in the neck was the presenting admission symptom in the 112 patients (100%), among which 60 patients had a solid thyroid nodule in the right lobe, 35 patients had a solid thyroid nodule in the left lobe, 16 patients had multiple thyroid nodules, and only one patient had a solid nodule in the thyroid isthmus. There were no significant differences in the malignancy rate due to nodule localization, but the number of multiple thyroid nodules was slightly higher in the low-risk group.

We identified an increased risk of thyroid carcinoma due to a positive family history in 20 patients (17.86%). A total of 17 patients had a history of thyroid diseases: Hashimoto thyroiditis in 15 patients (13.39%) and Graves’ disease in three patients (2.68%). Two patients (1.78%) had a history of neuroblastoma and had been treated with radiotherapy to the neck. There was no significant association between a positive history of radiotherapy (*p* = 0.117), a positive family history (*p* = 0.11), and autoimmune thyroid diseases (AITDs) (*p* = 0.508) with Graves’ disease (*p* = 0.278) or Hashimoto thyroiditis (*p* = 0.936).

We observed 59 children with a low risk of thyroid carcinoma without surgical intervention, which were named the low-risk group. Patients with a low risk of thyroid carcinoma were followed by ultrasonography and laboratory tests for 6–60 months.

The mean age of all patients was 13.4 ± 4.5 years (0.6–18 years). The mean age of the high-and low-risk groups was 14.7 ± 3.0 years (2.8–17.9 years) and 12.5 ± 5.2 years (0.6–8.1 years), respectively. The malignancy risk was slightly higher in older children (*p* = 0.047) (Figure 2).

There was no significant association with patient sex. Most patients were girls in both the low-and high-risk groups (Figure 3).

Thyroid ultrasonography was performed on all patients. The mean nodule volume size of the high-risk group and the low-risk group was 4.7 ± 6.5 mL and 3.5 ± 6.5 mL, respectively. The mean nodule volume was not significantly larger in the high-risk group (*p* = 0.1456).

In the low- and high-risk groups, the mean height, length, and width of the nodules were measured: 9.9 ± 8.3 and 14.1 ± 8.0, 13.7 ± 10.5 and 17.6 ± 10.1, and 15.1 ± 12.4 and 19.0 ± 14.8, respectively.

The malignancy risk was higher in children with a greater nodule height (*p* = 0.018) (Table 2).

Analysis of the ROC curves of the length, height, width, and volume of nodules in patients indicated that the pivotal dimension was the height of the nodule. In children whose histopathological examination revealed a malignant lesion of the thyroid, a statistically significantly greater nodule height was found compared to benign lesions. Nodules longer than 7 mm were statistically significantly diagnosed as thyroid cancer (Figure 4).

The hypoechogenicity of nodules or focuses was noted in 78.6% patients with thyroid cancer in ultrasound imaging and was significantly correlated with a final diagnosis (*p* = 0.010). The comparison of hypo- and hyperechogenic ultrasound of nodules is presented in Figure 5.

We did not observe any statistically significant correlation between increased blood flow in the vessels of the nodules and final cancer diagnosis in the histopathological examination. A comparison of blood flow in power Doppler ultrasonography is presented in Figure 6.

Microcalcifications on ultrasonography (<1 mm) were observed only in three patients in the high-risk group, finally diagnosed as thyroid papillary carcinoma, and in four patients, diagnosed as benign follicular nodules.

In 67 patients (59.82%), FNAB was performed. In 47 patients (70%), the FNAB result was benign (TBSRTC class II); in three patients (4.47%), non-diagnostic TBSRTC class I; in nine patients (13.43%), suspicious (TBSRTC class III); in eight patients (11.94%), malignant (TBSRTC class IV). For three patients (4.5%), the result was TBSRTC class V; TBSRTC class VI, for one patient (1.5%) (Figure 7).

A surgical procedure was performed in 37 patients (33.04%). A total of 12 patients (10.71%) underwent local excision of the suspected nodules, lobectomy was performed in 17 patients (15.17%), and TT was performed in eight children (7.14%). In three children (2.68%) with a high risk of thyroid carcinoma and a suspicion of metastases in the lymph nodes, frozen section examination was performed of the lymph nodes.

After histopathological nodule examination, the cases were benign lesions in 23 patients (62.16%) and malignant in 14 children (37.84%). Solitary benign thyroid nodules were identified in 16 children (40%). Multinodular goiter was diagnosed in three patients (7.5%); two children (5%) had Hashimoto thyroiditis or Graves’ disease. Of the malignant, the most common were papillary carcinoma (nine patients), follicular carcinoma (three children), and poorly differentiated thyroid carcinoma (one patient) (Table 3).

The sensitivity of the FNAB was 67% and the specificity was 90.2%.

Some interesting data concern the frequency of diagnosis of thyroid cancer, in particular, Bethesda classes, according to the histopathological examination. Category III TBSRTC is notable, usually in the case of adult patients classified for clinical observation; in our population, out of nine biopsies performed, thyroid cancer was detected in four cases (44%) (Table 4 and Figure 8).

For each TBSRTC class, the relative risk of developing thyroid cancer was also calculated by comparing the incidence of thyroid cancer in categories III, IV, and V in combination with VI with the combined incidence in classes I and II.

For classes I and II, a total of 50 thyroid biopsies were performed, 3 and 47, respectively, with two thyroid cancer results. The incidence of thyroid cancer in class III TBSRTC was 44% (four patients), so the relative risk of developing thyroid cancer for class III was 11.1 times higher than that for classes I and II combined. The incidence of thyroid cancer in class IV TBSRTC was 33% (three patients), so the relative risk of developing thyroid cancer for class IV was 8.3 times higher than that for classes I and II combined. The incidence of thyroid cancer in TBSRTC combined classes V and VI was one, so the relative risk of thyroid cancer for classes V and VI combined was 25 times higher than for classes I and II combined.

## 4. Discussion

Thyroid nodules are rare in the pediatric population, but often present a diagnostic and therapeutic challenge. According to the ATA guidelines, the Bethesda classification, which provides the basis for the development of treatment in adults, can also be used in children. However, according to our studies, there are significant differences that can have a considerable impact on the rapid and accurate diagnosis of pediatric patients. For this reason, in our work, we investigated the risk factors of thyroid cancer in children with thyroid nodules.

It is reported that those of female sex are at a higher risk of a nodular goiter unrelated to radiation sickness, but no significant relationship between the disease and the sex of the patients was found among the children studied [9]. Nevertheless, a large part of the study group was girls, who showed both a low and high risk of the disease.

The thyroid gland in children is sensitive to radiation, which often causes nodules [2]. In our study, it was difficult to clearly define the risk of disease associated with exposure to radiation because only two patients had undergone radiation therapy due to previous cancer.

A more frequent occurrence of nodular goiter has also been observed in patients with a positive family history, Hashimoto’s disease, and other thyroid diseases [3,4,5,6]. However, the analysis of the study group did not show a significant relationship between these risk factors and thyroid tumors.

Guidelines for the ultrasound diagnosis of thyroid nodules in adults are established and widely used [26,34,35,36]. However, it is debated whether the same guidelines can be applied to the pediatric population to effectively differentiate between benign and malignant lesions.

Over the last decade, differences in the risk stratification of thyroid nodules in pediatric ultrasound have been noted, which has resulted in the issuance of guidelines specifically for the pediatric population, such as the Assessment of the American College of Radiology (ACR) Thyroid Imaging Reporting and Data System (TI-RADS) and the update of the ATA guidelines [5,37,38]. Polat et al. analyzed the TI-RADS classification, and the results indicated its use is safe in children. However, the authors noted the small sample size and indicated that more research is needed on a larger group of children [39]. Regrettably, some other reports contradicted this conclusion and showed that the use of these guidelines in children carries a risk of missing cancer in up to 22% of pediatric patients [38,40,41]. The 2009 ATA guidelines recommended that the diagnostic criteria and treatment procedures for adults should also be used for the pediatric population [8].

Features of a focal thyroid lesion, such as being longer than wide, irregular margins or microcalcifications, always attract the examiner’s attention and are an indisputable indication for a biopsy. However, regarding the remaining features detected in adult patients, doubts have recently been expressed about their utility in assessing the risk of nodular malignancy in children.

One of the questionable criteria is the size of the nodule. In adults, the nodule’s size is not associated with increased malignancy risk, unlike in children. According to the 2018 Recommendation of Polish Scientific Societies *Diagnostics and treatment of thyroid cancer*, which coincide with the 2009 ATA guidelines for adults, changes below 1 cm in diameter visible on ultrasound do not require referral for FNAB, unless clinical examination or ultrasound finds features of an increased risk of malignancy, *RET* mutation, high levels of calcitonin, or no thyroid cancer metastases in a location other than the thyroid gland. However, lesions greater than 1 cm in diameter or detection of at least one of the previously mentioned criteria should be checked by FNAB [29]. Nonetheless, the criterion of the size of thyroid nodules in children is problematic because the size of the thyroid gland changes with age and the diameter of the nodule is not closely correlated with its potentially malignant nature. Therefore, guidelines specifically related to children were updated and announced in 2015 [5]. According to these guidelines, in children qualified for FNAB, the characteristic features in the ultrasound image and the clinical symptoms presented by the patient should be considered more than the size of the nodule. In our study, we found that children with a confirmed malignant nature of the lesion in histopathological examination had a greater height of the nodule. The cut-off point was 7 mm, above which the risk of thyroid cancer was significantly greater. Suh et al. did not find a correlation with nodule size, whereas Gannon et al. reported an increased risk of neoplasm with a size of ≥1 cm [1,42]. Richman et al. detected that the malignancy rate increased with increasing nodule size, but without a cut-off point [24].

The hypoechoic nature of the lesion is another controversial parameter among the authors of related publications around the world. In our work, hypoechoic changes in US correlated with an increased risk of malignant lesion diagnosis (*p* = 0.010). In the group of children diagnosed with cancer, this change occurred more often than in the group of children diagnosed with benign changes. The lesion was hypoechoic in 78.6% of children diagnosed with thyroid cancer. However, it cannot be ignored that 41.8% of the hypoechoic lesions in the US study were later found to be benign. Our data were in the range of 30–55% for benign lesions and 71–92% for malignant lesions obtained in a previous study [23]. The discrepancy in the sensitivity of this parameter between studies is significant. It varies between 26% and 41% and between 50.9% and 75.9%, suggesting that the importance of hypoechogenicity as a criterion for the malignancy of the lesion should be diminished, but it cannot be completely ignored [19,24,25,38].

Increased tumor vascularization is a recognized risk factor of malignancy and an indication for further diagnosis [5,43]. Unfortunately, several new findings have been published that contradict this approach [1,24,42,44]. Lingam et al. found that central or mixed color Doppler flow pattern and significant hypoechogenicity lead mostly to false-positive results [44]. In our children group, no statistically significant relationship was found between the presence of increased blood flow in the vessels and the diagnosis of thyroid cancer (*p* = 0.092), in line with new literature data. These conclusions do not allow for unequivocal differentiation of malignant lesions on the basis of vascularization, although the US Doppler test still remains useful in the clinical context, for example, to prevent damage to the vascular cluster during FNAB or to differentiate focal lesions from a cyst or blood clot [45].

Despite many attempts, so far, no one has managed to clearly establish the characteristics of malignant thyroid lesions in the pediatric population that we could safely use in clinical practice. We believe that it is necessary to conduct a large study, involving cooperation with other research centers.

Perhaps training in the performance of elastography will contribute to the wider dissemination of this examination and its inclusion in the permanent diagnostic algorithm of thyroid nodules in children. So far, little such research has been conducted, although the conclusions that elastography is a suitable complementary study and the high elasticity of a nodule in elastography is associated with a low risk of thyroid cancer are encouraging [46,47].

Fine needle aspiration biopsy was reported to have a very high sensitivity of 95%, a specificity of 86%, and an accuracy of 90%; however, in our study, they were lower than in other studies (sensitivity of 67% and specificity of 90.2%) [6,48]. FNAB is also a relatively cheap test. This examination is ideal for assessing thyroid nodules because of its properties and the use of TBSRTC.

TBSRTC has been widely used in the reporting of thyroid cytopathology and has allowed the nomenclature to be standardized around the world. The Bethesda classification has also facilitated therapeutic decisions, allowing for a more reliable comparison of statistical data, which improved the therapy of thyroid tumors.

Since the introduction of the TBSRTC in 2009, it has undergone some modifications to improve the system, e.g., reclassification of the noninvasive encapsulated follicular variant of papillary thyroid carcinoma (EFVPTC) in noninvasive follicular thyroid neoplasm with papillary-like nuclear features (NIFTP), which was aimed at reducing ROM and surgical overtreatment of benign lesions [49,50,51].

Sauter et al. focused on the problem of the overdiagnosis and overtreatment of benign thyroid nodules, while emphasizing the difficulties faced by clinicians in assessing the risk of Bethesda class III and IV malignancies. Using long-term clinical follow-up data, they showed more accurate positive and negative predictive values of diagnostic categories, reducing the overdiagnosis of AUS/FLUS, but maintaining high efficiency in the diagnosis of malignant and benign tumors. Thus, they indicated the role of pathologists who, by limiting the use of intermediate Bethesda classification categories (AUS/FLUS), if possible, could facilitate therapeutic decision making for clinicians [51].

The risk of malignancy of tumors in classes III, V, and VI has been confirmed by numerous studies, and the choice of clinical procedure is not difficult, e.g., ATA guidelines can be used. Benign lesions (class II) are further assessed with US and FNAB, and molecular tests are performed; only in some cases is surgical treatment used. In the case of lesions highly suspected of being malignant or confirmed malignant (classes V and VI), surgery is the treatment of choice [5,33,36,52]. Problems with treatment choice for the intermediate classes of TBSRTC (classes III and IV) has already been addressed in several studies [49,52,53].

Both classes III and IV have an intermediate risk of malignancy: 6–30% and 10–40%, respectively [33]. The risk is so high that choosing the right therapy tailored to a specific patient is difficult and can easily be affected by errors. All these problems require work to adapt the Bethesda system to assess children with thyroid tumors, which can be achieved by planning and analyzing statistical studies in the pediatric population.

According to our observations, Bethesda system class III in children may already be a risk factor of malignant thyroid neoplasm, whereas in adults, the risk of malignancy increases significantly in Bethesda class IV, as reported in other studies [1,2,6,10]. Sources report that the risk of malignancy in classes V and VI is 100%, which is confirmed by our results [10].

Regarding adult patients with undefined preoperative thyroid cytopathology, surgery is only performed after re-biopsy or molecular tests. In children, the guidelines are more radical, and surgery is recommended after the first biopsy [10,54].

The more aggressive treatments in children are due to their longer life expectancy. In children, genetic predisposition to the development of thyroid neoplasms is more common. One must also consider the increased influence of growth factors, especially increased insulin-like growth factor-1 (IGF-1) in childhood and adolescence. The IGF complex seems to play an important role in thyroid cancer. The expression of the IGF-1 system is enhanced in thyroid cancer (especially in PTC) compared to other thyroid diseases [55].

## 5. Conclusions

The thyroid cancer risk was significantly increased in children with a nodule height greater than 7 mm and with a hypoechogenic structure in ultrasonography.

TBSRTC is effective for the qualification of children for surgery of the thyroid. Approximately 25% of FNAB in children with nodular goiter were qualified to classes III and VI of the TBSRTC. Class III TBSRTC was connected in our pediatric with a higher risk of thyroid cancer in comparison to the adult population. Classes III and VI should be an indication for partial or total thyroidectomy because of the significantly high risk of thyroid cancer.

## Figures and Tables

**Figure 1 jcm-10-04455-f001:**
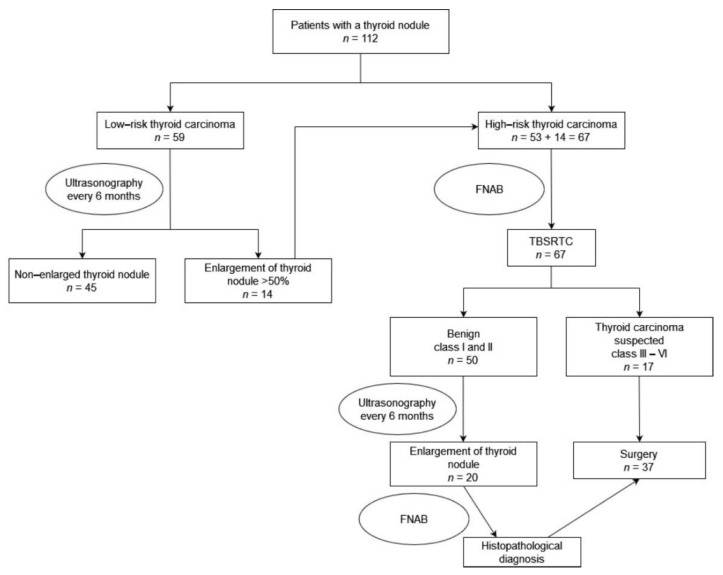
Diagnostic pathway for patients with thyroid nodules. FNAB- Fine needle aspiration biopsy; TBSRTC-The Bethesda System for Reporting. Thyroid Cytopathology.

**Figure 2 jcm-10-04455-f002:**
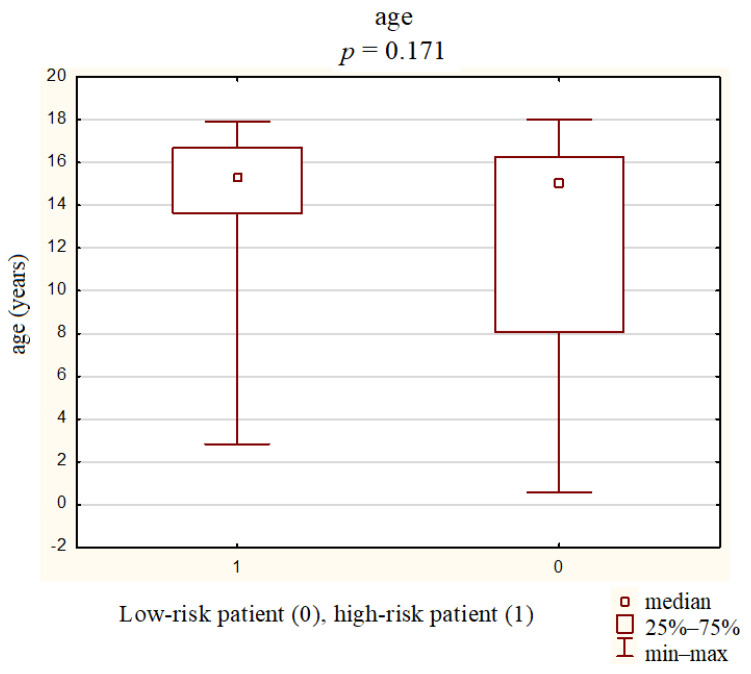
Mean age of the groups with a high risk and a low risk of thyroid cancer.

**Figure 3 jcm-10-04455-f003:**
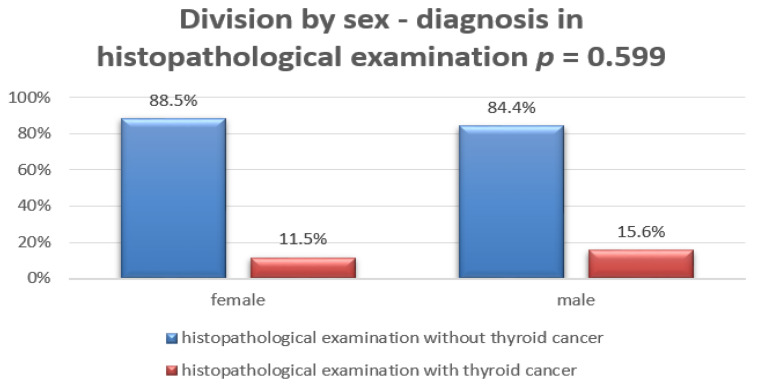
Division by sex: diagnosis based on histopathological examination.

**Figure 4 jcm-10-04455-f004:**
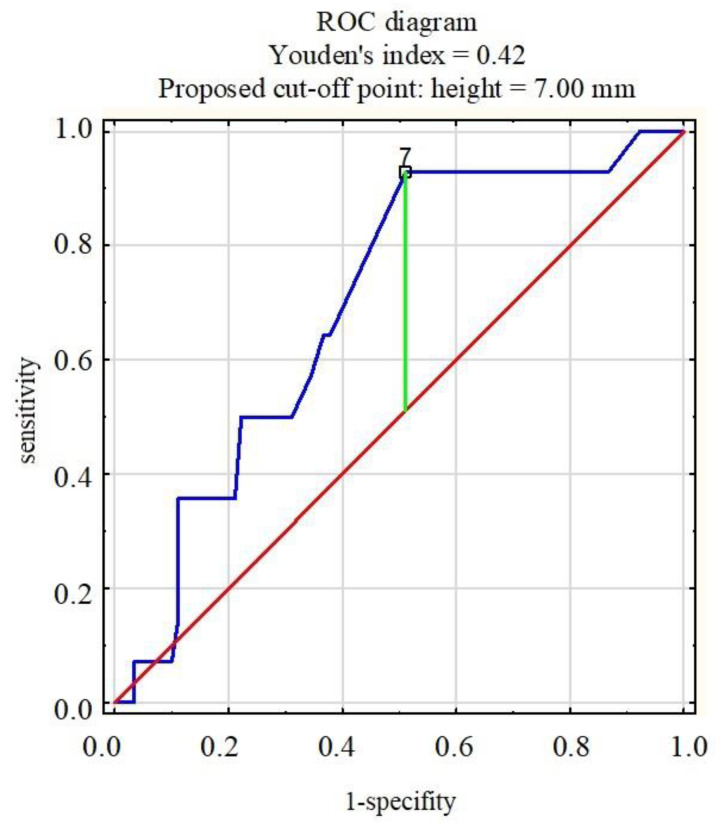
ROC diagram (receiver operating characteristic curve): nodule height and the risk of malignancy.

**Figure 5 jcm-10-04455-f005:**
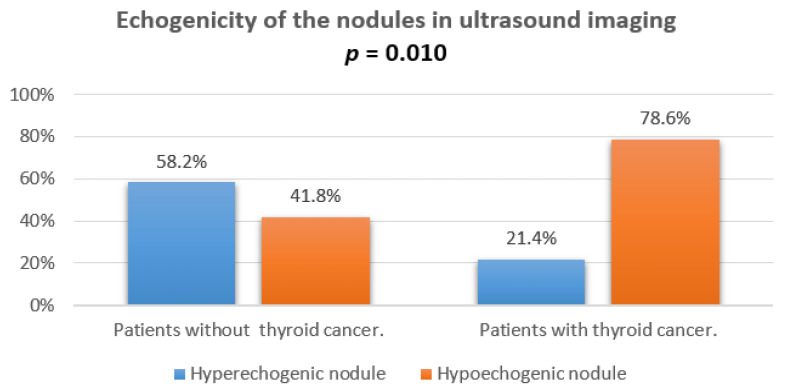
Echogenicity of the nodules in ultrasound imaging.

**Figure 6 jcm-10-04455-f006:**
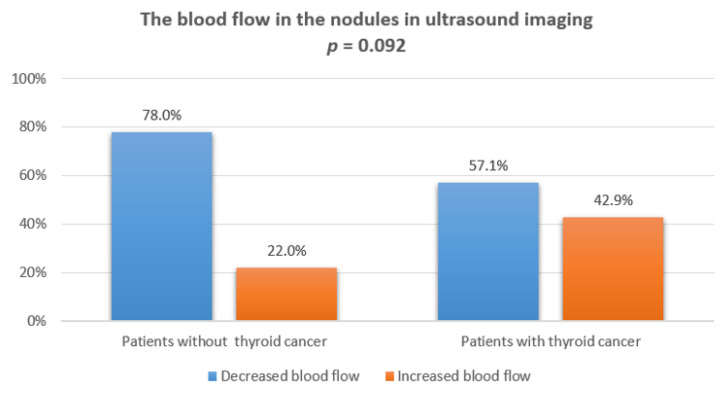
The blood flow in the nodules in ultrasound imaging.

**Figure 7 jcm-10-04455-f007:**
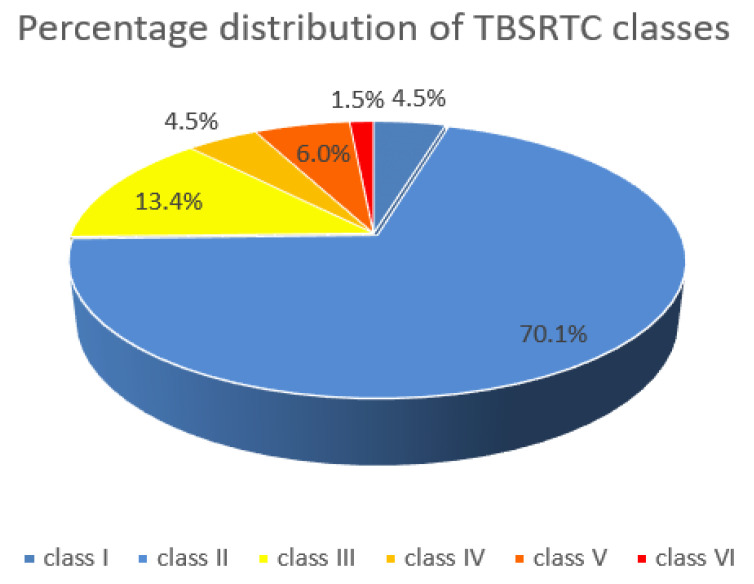
Percentage distribution of the TBSRTC (The Bethesda System for Reporting. Thyroid Cytopathology) classes.

**Figure 8 jcm-10-04455-f008:**
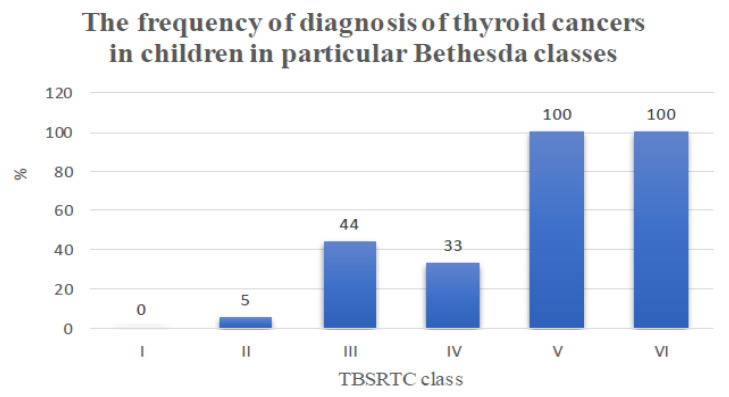
The frequency of diagnosis of thyroid cancers in children in particular Bethesda classes.

**Table 1 jcm-10-04455-t001:** Clinical and ultrasonographic features of increased risk of malignancy [21,22,23,24,25,26].

Clinical Features of Increased Risk of Malignancy	Ultrasonographic Features of Increased Risk of Malignancy
Lymph node metastasis or distant metastasis	Features indicating the possibility of metastasis of thyroid cancer to cervical lymph node microcalcifications, lithocystic character, hyperechogenicity, round shape, transverse dimension > 5 mm, lack of an echogenic fatty cavity, marginal or chaotic vascularization of lymph nodes
History of neck exposure to ionizing radiation	Features of infiltration of the thyroid capsule with or without infiltration of the surrounding organs
History of familial thyroid cancer (for medullary thyroid cancer)	Presence of microcalcifications in the focal lesion of the thyroid gland
Rapid growth of the nodule	Hypoechoic nature of a focal lesion
Hard nodule, attached to the surroundings	Shape of the focal lesion (longer than the width)
Diameter of the nodule > 4 cm	Diffuse margins
Appearance of a thyroid nodule before the age of 20 or after the age of 60 years	Features of increased, chaotic vascular flow centrally in the lesion
Paralysis of the laryngeal nerves, especially one-sided	Solid character of a focal lesion

**Table 2 jcm-10-04455-t002:** Dimensions of the thyroid nodules in patients without and with thyroid cancer (Mann–Whitney U-test, significance *p* < 0.05).

Nodule Dimensions	Patients without Thyroid Cancer Mean ± SD	Patients with Thyroid Cancer Mean ± SD	*p*
Length (mm)	13.7 ± 10.5	17.6 ± 10.1	0.090
Height (mm)	9.9 ± 8.3	14.1 ± 8.0	0.018
Width (mm)	15.1 ± 12.4	19.0 ± 14.8	0.296
Volume (mm^3^)	4.7 ± 6.5	3.5 ± 6.5	0.146

**Table 3 jcm-10-04455-t003:** The final histopathologic diagnoses in children after FNAB. The three children with class I and 29 children with class II TBSRTC did not undergo surgery of the thyroid. TBSRTC (The Bethesda System for Reporting. Thyroid Cytopathology); AITD (autoimmunity thyroid diseases).

TBSRTC	AITD (Hashimoto or Graves’ Disease)	Multinodular Goiter	Benign Follicular Nodule or Adenoma	Papillary Carcinoma	Follicular Carcinoma	Poorly Differentiated Thyroid Carcinoma
I	0	0	0	0	0	0
II	0	1	13	2	0	0
III	2	2	1	4	0	0
IV	0	0	2	1	0	0
V	0	0	0	1	3	0
VI	0	0	0	0	0	1

**Table 4 jcm-10-04455-t004:** The frequency of diagnosis of thyroid cancers in children in particular Bethesda classes.

TBSRTC Class	Number of Cancer Diagnoses of the Number of Biopsies Performed	Percentage of Children Diagnosed with Cancer (%)
I	0 out of 3	0
II	2 out of 47	4
III	4 out of 9	44
IV	1 out of 3	33
V	4 out of 4	100
VI	1 out of 1	100

## Data Availability

Data Availability Statements in section University Children Hospital in Lublin Poland.

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
