# Peer review of "Thyroid Cancer Risk Factors in Children with Thyroid Nodules: A One-Center Study"

_jcm, 2021, doi:10.3390/jcm10194455_

Round 1
Reviewer 1 Report
Dear authors,
with great interest, I read the manuscript “Thyroid cancer risk factors in children with thyroid nodules” submitted to the Journal of Clinical Medicine. The study evaluates the reik factors for thyroid cancer in children suffering from thyroid nodules in a study cohort of 112 children between 0.6 and 18 years of age.
The submitted manuscript is well designed and its methodology and statistical analysis are appropriate for its stated aim. The intention for the study is presented adequately and the results are clear-cut. The results of the submitted manuscript are interesting especially for endocrinologists, surgeons, radiologists and nuclear medicine physicians and therefore certainly worth publication. The results may also be a valuable help for the development of upcoming guidelines.
However, I have some revisions concerning the manuscript.
Introduction
The Introduction is rather long. I would suggest to shorten this section to improve readability.
Methods
This section is well written and describes the study collective and the used statistic methods adequately. The used statistically analysis is in my opinion appropriate for its stated aim.
Results
The results are presented adequately.
Discussion
This section is again well written and clearly to the point. However, this section is again rather long and would benefit when shortened.
Conclusion
The Conclusion however is really short and should be extended by naming the main results of the analysis.
Tables
The tables are appropriate.
One remark to table 4: Please replace “…had not surgery…” by “… had no surgery…”
Figures
Some figures should be revised:
Figure 3: please do not indicate decimal places when scaling the y-axis.
Figure 5: please do not indicate decimal places when scaling the y-axis.
Figure 6: please replace “Patietns” by “Patients”
Figure 6: please do not indicate decimal places when scaling the y-axis.
Figure 6: please replace “Patietns” by “Patients”
Grammar and style
The English used in the manuscript is not adequate and should be proofread by a medical writer, an English editing service or at least a native speaker. Here are some examples:
“In all patient were determined calcitonin level in serum”
“The most patients was female…”
“Mean nodule volume was no significantly larger…”
“…of the nodules were measured…”
“Malignancy risk was higher in children with…”
“Nodules over 7mm statistically significantly was diagnosed as…”
“…focuses was noticed in 78.6% patients with …”
“After histopathological nodules examination…”
…
General
Please provide a line numbering when submitting your revision. This will make it a lot easier for the reviewer.
Author Response
Thank you to for your valuable advice and remarks. We have decided to make the changes according to your suggestions.
We removed selected fragments of the introduction and discussion section in order to improve readability – we marked up these fragments using the “Track Changes” function, such that can be easily viewed.
The conclusions summarize the unambiguous results we have obtained. The remaining conclusions are uncertain and require further research, i.a. due to too small population group.
We corrected the error in table 4 and in figure 3, 5 and 6. Due to the deletion of one of the tables, subsequent numbering has also been corrected.
Moreover, the whole work has been moved to the template, where the number of lines is marked.
In connection with the changes made - the order of references has also been corrected.
In order to improve the English language our work will be sent for linguistic proofreading to MDPI's English editing service.
Thank you to for your valuable advice and remarks. We have decided to make the changes according to your suggestions.

Reviewer 2 Report
This is a manuscript on thyroid cancer risk factors of thyroid nodules in young generations.
Critically, the introduction is too long and the aim of this study is obscured. Try describing background only related to the aim of the current study.
This is minor, but the Chernobyl accident and Fukushima issue does not seems to be ‘similar’ especially with regard to radiation.
The authors should revise the volume and sections to be concise.
Author Response
Thank you to for your valuable advice and remarks. We have decided to make the changes according to your suggestions.
We removed selected fragments of the introduction (including a fragment about the Chernobyl and Fukushima accident in order to improve readability – we marked up these fragments using the “Track Changes” function, such that can be easily viewed.
In connection with the changes made - the order of references has also been corrected.
In order to improve the English language our work will be sent for linguistic proofreading to MDPI's English editing service.

This manuscript is a resubmission of an earlier submission. The following is a list of the peer review reports and author responses from that submission.